# Alpha-Synuclein mRNA Level Found Dependent on L444P Variant in Carriers and Gaucher Disease Patients on Enzyme Replacement Therapy

**DOI:** 10.3390/biom13040644

**Published:** 2023-04-03

**Authors:** Paweł Dubiela, Paulina Szymańska-Rożek, Andrzej Eljaszewicz, Patryk Lipiński, Piotr Hasiński, Dorota Giersz, Alicja Walewska, Marlena Tynecka, Marcin Moniuszko, Anna Tylki-Szymańska

**Affiliations:** 1Department of Regenerative Medicine and Immune Regulation, Medical University of Bialystok, 15-269 Bialystok, Poland; 2Department of Pathophysiology and Allergy Research, Medical University of Vienna, 1090 Vienna, Austria; 3Faculty of Mathematics, Informatics and Mechanics, University of Warsaw, 02-097 Warsaw, Poland; 4Department of Pediatrics, Nutrition and Metabolic Diseases, The Children’s Memorial Health Institute, 04-730 Warsaw, Poland; 5Department of Internal Medicine and Gastroenterology, Municipal Hospital, 43-100 Tychy, Poland

**Keywords:** Gaucher disease, Parkinson’s disease, glucosylsphingosine, α-synuclein, enzyme replacement therapy

## Abstract

Gaucher disease (GD) is the most frequent sphingolipidosis, caused by biallelic pathogenic variants in the *GBA1* gene encoding for β-glucocerebrosidase (GCase, E.C. 3.2.1.45). The condition is characterized by hepatosplenomegaly, hematological abnormalities, and bone disease in both non-neuronopathic type 1 (GD1) and neuronopathic type 3 (GD3). Interestingly, *GBA1* variants were found to be one of the most important risk factors for the development of Parkinson’s disease (PD) in GD1 patients. We performed a comprehensive study regarding the two most disease-specific biomarkers, glucosylsphingosine (Lyso-Gb1) and α-synuclein for GD and PD, respectively. A total of 65 patients with GD treated with ERT (47 GD1 patients and 18 GD3 patients), 19 *GBA1* pathogenic variant carriers (including 10 L444P carriers), and 16 healthy subjects were involved in the study. Lyso-Gb1 was assessed by dried blood spot testing. The level of α-synuclein as an mRNA transcript, total, and oligomer protein concentration were measured with real-time PCR and ELISA, respectively. α-synuclein mRNA level was found significantly elevated in GD3 patients and L444P carriers. GD1 patients, along with *GBA1* carriers of an unknown or unconfirmed variant, as well as healthy controls, have the same low level of α-synuclein mRNA. There was no correlation found between the level of α-synuclein mRNA and age in GD patients treated with ERT, whereas there was a positive correlation in L444P carriers.

## 1. Introduction

Gaucher disease (GD) is the most frequent sphingolipidosis, caused by biallelic pathogenic variants in the *GBA1* gene. More than 400 *GBA1* pathogenic variants have been described so far, some of them, such as c.1226A>G (p.Asn409Ser; N370S), c.1448T>C (p.Leu483Pro; L444P), c.84dup, c.115+1G>A (IVS2+1G>A) being responsible for 90% of cases among Ashkenazi Jews population and 60% of non-Ashkenazi patients [1,2,3]. *GBA1* defects lead to β-glucocerebrosidase (GCase) enzyme deficiency, which results in accumulation of its substrate, glucosylceramide (GlcCer, Gb1), mostly in the monocyte-macrophage system [4]. The disease course is clinically variable and depends on specific *GBA1* variants. Three clinical forms have been identified. The most common form, type 1 (GD1), is believed not to cause neurological damage, whereas neuronopathic forms, type 2 (GD2) and 3 (GD3), are characterized by progressive neurological impairment [5].

Glucosylsphingosine (Lyso-Gb1) is a deacylated form of glucosylceramide. In GD patients, Lyso-Gb1 plasma concentrations are found to be markedly increased above the cut-off level (12–14 ng/mL), compared to healthy individuals [6]. The association between patients genotypes and Lyso-Gb1 levels has also been reported. GD patients homozygous for the L444P variant had higher Lyso-Gb1 levels (median, 184.5 ng/mL) than those with the N370S variant on at least one *GBA1* allele (median, 143.1 ng/mL) [7]. Moreover, the brain levels of Lyso-Gb1 were shown to be 22–51 times and 38–694 times higher in patients with GD3 and GD2, respectively, in comparison to healthy individuals. At the same time, the brain levels of Lyso-Gb1 in GD1 patients were found within the normal range and were comparable to those in healthy subjects, thus suggesting that elevated Lyso-Gb1 levels may contribute to the neurological manifestations characteristic of neuronopathic GD [8]. Therefore, it is not surprising that Lyso-Gb1 is believed to be the most reliable biomarker currently available for the diagnosis, prognosis, and treatment monitoring of GD patients, as reviewed by Revel-Vilk et al. [9].

Parkinson’s disease (PD) is the second most common neurodegenerative disorder, with an average age of 62 years at diagnosis and a risk of about 3% to 4% in developed countries [10,11]. Aside from the involvement of several commonly known neurotransmitters, including the cholinergic, noradrenergic, and serotonergic pathways [12,13,14,15], there is growing evidence of the genetic background [16]. The formation of intracellular deposits of proteins and lipids that precede neuronal damage is believed to be the most critical factor responsible for PD development. The main players in this pathology are the Lewy bodies, with deposits of α-synuclein (α-SNCA), whose robust presence was found in neurodegenerative structures [17,18,19,20,21]. It is in line with the fact that *SNCA* pathogenic variants are commonly associated with PD [22]. Those two discoveries led to the pathological staging of the disease proposed by Braak et al. [23]. Even now, the fact that α-SNCA deposition occurs early in PD is applied in immunohistochemical tests and is referred to as the gold standard in the neuropathological evaluation of PD [24].

The α-SNCA protein consists of 140 amino acid residues and reveals a sophisticated nature. Since in aqueous solutions it does not have a defined structure, it was referred to as a “natively unstructured protein” [25]. However, it has α-helical structures upon binding to negatively charged lipids, such as phospholipids present in the cellular membranes [26]. Unfolded (native) monomeric α-SNCA is able to aggregate into oligomeric species that can be stabilized by β-sheet-like interactions. Such oligomers can further transform into higher molecular weight, insoluble fibrils, seen in different synucleinopathies [27].

The association of *GBA1* variants with PD and dementia (*GBA1*-associated Parkinson’s disease) has been extensively studied over the past few years. Although GD1 is not associated with neurological impairment, there is an approximately 26-fold higher life-time risk of developing PD compared to the general population. Specifically, from 5% to 7% of GD1 patients may develop PD before the age of 70 and 9 to 12% before the age of 80, with a mean age of 57 years at the onset [28]. On the contrary, the glucocerobrosidase E326K variant is known to predispose to PD, but does not cause GD [29]. There is scarce information about GD3 patients and PD, due to the low number of diagnosed cases and reduced life expectancy [30]. Interestingly, carriers of *GBA1* pathogenic variants are also at a higher risk of developing PD [5,31,32]. Further studies are aimed at explaining this phenomenon. It was shown that Lyso-Gb1 correlated with elevated α-synuclein levels in neurons from patients with neuronopathic GD or PD and also promoted the formation of α-SNCA oligomers [33,34,35]. Although the available data provides some insight regarding the potential roles of Lyso-Gb1 and α-SNCA in the interplay between GD and PD, there are still puzzles missing from the picture.

Therefore, we aimed to provide comprehensive data and find possible interactions between *GBA1* pathogenic variants, including GD1 and GD3 patients and obligatory carriers from their families, Lyso-Gb1 concentration, and α-SNCA, by considering total protein concentration, oligomer concentration, and *SNCA* mRNA transcript level.

## 2. Materials and Methods

### 2.1. Patients and Blood Samples

A total of 65 patients with GD treated with ERT, 19 *GBA1* pathogenic variant carriers, and 16 healthy subjects were recruited to the study. The protocol of the study was approved by the local Ethical Committee of The Children’s Memorial Health Institute, Warsaw, Poland (number 51/KB/2019). A written, informed consent was obtained from all participants. The study was conducted in accordance with the ethical principles outlined in the Declaration of Helsinki.

Blood samples were obtained at one point in time from GD patients undergoing their annual clinical follow-up visit, combined with the biochemical analysis. In addition, family members and healthy subjects were invited to participate in the study. The clinical interview was standardized and performed by a single experienced clinician. Lyso-Gb1 levels in plasma were measured using high-pressure liquid chromatography–tandem mass spectrometry at Archimed Life Sciences Vienna, as previously described [7]. Genetic data were retrieved from the medical records of GD patients and acquired for healthy subjects at Archimed Life Sciences Vienna [7]. All the samples were blinded and sent for α-synuclein laboratory analysis in the coded format.

### 2.2. Biobanking

All the blood samples analyzed in the study were collected, transferred, and stored at the Medical University of Bialystok Biobank, Poland (member of BBMRI.pl, a national node of the European Biobank Network, BBMRI-ERIC), according to the standard operating procedure for biological materials. Briefly, the blood was centrifuged at room temperature for 5 min at 400× *g* to separate plasma. Next, plasma specimens were centrifuged at 4 °C for 5 min at 1200× *g* to remove residual cells. Aliquoted samples were biobanked at −80 °C in a controlled environment. Peripheral blood mononuclear cells (PBMCs) were isolated from plasma-depleted blood samples by means of density gradient centrifugation (Pancoll). Freshly isolated cells (1 million) were lysed using RLT buffer (Qiagen, Hilden, Germany) and biobanked at −80 °C. The remaining cells were cryopreserved in a freezing medium (Fetal Bovine Serum, PAA supplemented with DMSO, Sigma-Aldrich, St. Louis, MO, USA) and stored in LN_2_.

### 2.3. α-Synuclein mRNA Analysis

Total RNA was isolated from PBMC lysates using the #RNeasy Mini Kit isolation kit (Qiagen, Hilden, Germany) according to the manufacturer’s protocol. Quantity and quality of RNA assessments were performed using a UV/VIS spectrophotometer, the NanoDrop 2000c (Thermo Fisher Scientific, Inc., Waltham, MA, USA). The RNA was reverse transcribed using High-Capacity RNA-to-cDNA kit (Applied Biosystems, Waltham, MA, USA; Thermo Fisher Scientific, Inc.) according to the manufacturer’s instructions. *SNCA* Hs00240906 gene expression and housekeeping gene (*GAPDH* Hs02786624, Thermo Fisher Scientific) were analyzed with Taq probe (TaqMan^TM^ Universal Master Mix, with UNG, Applied Biosystems; Thermo Fisher Scientific, Inc.) using StepOne Plus Real-Time System (Applied Biosystems; Thermo Fisher Scientific, Inc.). Each sample was analyzed in triplicate.

### 2.4. α-Synuclein Total and Oligomer Protein Concentration

The total α-SNCA protein concentration was measured from the plasma with Human alpha-synuclein DuoSet ELISA (R & D Systems Europe, Ltd., Minneapolis, MN, USA) according to the manufacturer’s protocol. The detection range of the test is 0.157 ng/mL–10 ng/mL. For the samples exceeding the range, a signal threshold value of 0.157 ng/mL and 20 ng/mL was assumed. α-SNCA oligomer concentration was assessed in the plasma using a commercially available human alpha synuclein oligomer (A-SNCO) ELISA kit (MyBiosource.com), according to the manufacturer’s protocol. The detection range of the test is 0.625 ng/mL–20 ng/mL. The reaction plates were analyzed with an automated light absorption reader (Ledetec). The results were calculated according to the standard curve using MicroWin2000 software (Baton Rouge, LA, USA).

### 2.5. Statistical Analysis

The data were first checked for normality with the Shapiro–Wilk test. Since in the vast majority of groups the distributions were not normal, a proper variant of the t-test was used to assess statistical differences in distributions. “Raw” *p*-values of these comparisons are reported on the box-plots, if only they are less than 0.05. However, to minimize the probability of reporting insignificant results as significant, we adjust the *p*-values with the Benjamini–Hochberg correction and report as significant only those *p*-values that are less than the largest *p*-value that is less than its Benjamini–Hochberg critical value (see [36]). Finally, the data were investigated for age correlations in the five groups for all four markers. The coefficients of Pearson’s linear correlation were then assessed for statistical significance with t-tests. Statistical analysis was performed in RStudio, and figures were drawn in Gnuplot.

## 3. Results

### 3.1. Patients’ Characteristics

A total of 65 Polish patients with GD, 47 of type 1 and 18 of type 3, were enrolled in the study. All the patients displayed a decreased activity of β-glucocerebrosidase, combined with genetic confirmation (detection of *GBA1* biallelic pathogenic variants). Regarding patients’ genotype, the entire GD3 population consisted of L444P homozygotes, and GD1 patients presented various variants, see Table 1.

A total of 19 carriers of *GBA1* pathogenic variants and 16 healthy individuals served as comparators. Among the 19 *GBA1* carriers, 10 were identified as L444P variant carriers, and the remaining nine have either N370S or an unknown *GBA1* pathogenic variant. Most of the patients were adults, with one adolescent among the carriers and three adolescents in the group of GD1 patients. All the patients with GD were treated with ERT. Basic demographic data are presented in Table 2. Due to the rarity and nature of the disease, it was not possible to match patients in terms of age, so GD3 patients were significantly younger.

### 3.2. Lyso-Gb1 Concentration

According to the threshold in the Archimed Life Science Lab, Lyso-Gb1 levels less than 14 ng/mL were considered normal. All patients diagnosed with GD had Lyso-Gb1 levels higher than this threshold. GD1 and GD3 patients had median values of 71.5 ng/mL and 64.6 ng/mL (*p* > 0.05), respectively. The median biomarker concentration was significantly higher in GD patients than in *GBA1* variant carriers (median of 7.51 ng/mL, *p* < 10^−9^) and healthy control subjects (8.49 ng/mL, *p* < 10^−8^). There was no statistically significant difference between the two healthy cohorts. Elevated Lyso-Gb1 levels were found in one healthy subject and seven *GBA1* variant carriers (Figure 1, Table 2).

### 3.3. SNCA Gene Expression

The levels of *SNCA* mRNA transcript were substantially higher in GD3 patients and the L444P carriers (with medians of 77.48 and 66.42 (2^−ΔcT^), respectively). Healthy controls, as well as GD1 patients and N370S carriers, had very low levels of *SNCA* mRNA transcript, with medians of 0.83, 047, and 0.52 (2^−ΔcT^), respectively (Figure 2, Table 2). The difference in *SNCA* mRNA transcript level between carriers of L444P on at least one *GBA1* allele and any other tested group was found to be statistically significant. Figure 2 is the only figure where GD patients were divided into GD1 and GD3 subgroups and carriers were divided into “L444P mutation carriers” and “unknown mutation carriers”. This is because of the significantly different results obtained for the pairs of subgroups: GD1/GD3, and “L444P mutation carrier”/“unknown mutation carrier”. Note that all other markers analyzed were statistically equal for these two pairs of subgroups.

### 3.4. α-SNCA Total Protein Level

The median total α-SNCA protein concentration was found to be the highest in the control group (1.34 ng/mL). Both GD patients and *GBA1* carriers had medians of 0.16 and 0.54 ng/mL, respectively (Figure 3, Table 2). The difference between GD patients and the control group appeared to be statistically significant (*p*-value of 0.0075), but since we had to account for multiple comparisons, this *p*-value after the adjustment was not assessed as significant.

### 3.5. α-SNCA Oligomer Concentration

There were no statistically significant differences in the levels of oligomer α-synuclein between GD patients, *GBA1* carriers, and the control group. The medians were as follows: 1.81, 1.69, and 2.08 ng/mL, respectively (Figure 4, Table 2).

### 3.6. Correlations with Age

The measured factors, namely: Lyso-Gb1, *SNCA* transcript level, α-SNCA protein, and oligomer concentration, were checked for linear correlation with age in five groups: GD1, GD3, unknown *GBA1* variant carriers, L444P carriers, and the healthy control. The only correlations found were as follows: total α-SNCA protein level among GD3 patients (r = −0.52, *p* = 0.029), *SNCA* mRNA level among the carriers of an unknown mutation (r = −0.6, *p* = 0.19), and *SNCA* mRNA level in L444P mutation carriers (r = 0.74, *p* = 0.039) (Appendix A). All other variables in the tested cohorts were found to be uncorrelated with age. The results of the whole analysis are presented in Appendix A. 

## 4. Discussion

Since the *GBA1* pathogenic variants are the best described risk factors for Parkinson’s disease [5,31,32], we aimed to understand the molecular level relevance of the most commonly reported *GBA1* variants, N370S and L444P, and performed the study on Lyso-Gb1 and α-synuclein as the most relevant markers of Gaucher and Parkinson’s disease, respectively. We found elevated mRNA *SNCA* level among GD3 patients and L444P mutation carriers compared to other tested cohorts, with no concomitant elevation in α-synuclein protein level or increase in its oligomerization.

The role of Lyso-Gb1 as the diagnostic marker in GD patients’ management is well established [9]. Interestingly, no statistically significant difference in Lyso-Gb1 levels was found among treated patients with GD1 and GD3. This observation can be somehow surprising since the Lyso-Gb1 level reflects the severity of the disease [7]. Based on the recently reported studies, it was shown that Lyso-Gb1 is useful for treatment monitoring [37]. There was a clear correlation between certain genotypes and the patients who were naïve to therapy, especially with GD3, which had the highest biomarker concentrations [37]. On the other hand, as shown by Dekker et al., only among N370S homozygotes was there a clear correlation between disease severity and plasma Lyso-Gb1 concentrations [6]. The reported outcomes show differences caused not only by genotypic variations among the study groups but also reflect the ERT impact. It has recently been shown by our group that Lyso-Gb1 correlates with the response to the treatment measured by chitotriosidase activity only in some ranges of dosing or for splenectomized patients. The coefficient of determination, R^2^, follows a trend, increasing steadily with dosing, then suddenly dropping to 0.16 for patients treated with high doses (>35 U/kg bm/every other week). Therefore, one of the strengths and limitations of our study is its intrinsic heterogeneity. Patients were treated with different doses of ERT (15–60 U/kg, on average 30 U/kg bm/every other week). Another interesting finding is the elevated level of Lyso-Gb1 among L444P carriers. Individuals from the cohort exceeded the threshold level (14 ng/mL) and in one of the cases described in this paper, the concentration was more than three times higher (47 ng/mL).

One of the revolutions of the last decade was the finding of *GBA1* pathogenic variants as the most important genetic predisposing risk factor for PD development [38,39]. At the same time, the role of α-SNCA in the etiopathogenesis of PD was well characterized, but there was no clear link on how *GBA1* defects can be a risk factor for synucleinopathies. The link between *GBA1* pathogenic variants (e.g., N370S, L444P) and the accumulation of α-synuclein and formation of its aggregates can be explained by dysfunctions of proteolysis machinery due to ER stress and defects of chaperone-mediated autophagy [40]. Loss of GCase function leads to accumulation of lipids such as Gb1 and Lyso-Gb1, which directly interact with and stabilize α-synuclein oligomers in the lysosomal compartment. The consequent α-synuclein accumulation inhibits ER-to-Golgi GCase trafficking but also causes general lysosomal dysfunction, defective mitophagic and autophagic clearance pathways, and Ca^2+^ homeostasis dysfunction [40].

Our data provide some interesting insights. There was no increase in the concentration of α-synuclein oligomers among GD patients in comparison to healthy controls and *GBA1* variant carriers. These results are contrary to the previous study showing an increased dimerization of α-synuclein in the erythrocytes of GD patients [41]. The results should be compared with caution since the authors narrowed the conclusions to the analysis of red blood cells as they were not able to detect α-synuclein in any other tissue samples. In another study, by Pchelina et al., there was found to be an increased concentration of α-SNCA in GD patients compared to the control group. However, this difference was found not to be specific for GD patients but also presented for other lysosomal storage disorders. The authors also found the effect of the ERT duration on the total amount of α-SNCA and the age to be negatively correlated with the protein oligomerization [42].

The second part of our approach aimed to identify differences in the total concentration of α-synuclein. We found slightly elevated levels of total α-SNCA protein among healthy controls in comparison with GD patients; however, in our statistical model, the difference was not significant. There are not enough literature data to refer our observations to. The results from the mice model showed an increase in α-SNCA concentration in the neurons of individuals with GCase deficiency. The authors suggested that the increase in protein concentration, and thus α-SNCA polymerization, leads to neurotoxicity [43]. Another study performed on samples from cerebrospinal fluid among *GBA1* pathogenic variant carriers revealed the opposite results, with decreased α-SNCA levels [44]. Our data, obtained from blood samples, suggest that there is a moderately negative correlation between total α-SNCA concentration and age among GD3 patients. These results partially support the observation made by Lerche et al. and Pchelina et al. indicating that with time and duration of ERT treatment, the level of α-SNCA concentration decreases and protein oligomerization is not promoted [42,44]. Perhaps this could be the case in our study, where the majority of GD3 patients in our cohorts were diagnosed and started treatment very early, continuing to receive ERT for over 20 years (data not shown).

The most interesting observations are coming from the results showing the elevated mRNA *SNCA* level among GD3 patients and L444P carriers compared to other tested cohorts.

The elevated mRNA *SNCA* level was also previously reported by Chiba-Falek et al. in samples from the brains of sporadic PD patients [45]. The authors concluded that the obtained results suggest that elevated expression levels of SNCA-mRNA are found in the affected regions of the PD brain and support the hypothesis that increases in α-synuclein expression are associated, among other factors, with the development of sporadic PD.

L444P variant has been previously associated with PD development in various ethnic groups [46,47,48,49,50,51,52,53,54,55,56,57]. However, Becker-Cohen et al. have recently reported a study on the comprehensive assessment of prodromal parkinsonian features in 98 carriers (including 71 carriers of the N370S variant) of GD. L444P variant carriers exhibited the highest percentage of abnormal clinical tests assessing various domains of PD compared to N370S variant carriers [57]. However, there are some contradictory results on the presence of this pathogenic variant in early or late forms of PD [47,58,59]. Our data showed not only an elevated mRNA concentration but also a positive correlation of this factor with age. This, in turn, means that the *SNCA* expression level is increasing, but this does not correspond with either the total or oligomer α-SNCA concentration level. There is the possibility that protein is stored in the cells and therefore was not caught by our methods, as suggested by Barbour et al. [60]. Another hypothesis could be that the translation process is disrupted by proteases/regulators. It was shown that α-synuclein binds its own mRNA and prevents the initiation of translation at least to some extent [61,62].

When we add the results from unknown *GBA1* variant carriers showing a negative correlation with age, i.e., a decrease in the mRNA concentration with aging, we can speculate why some researchers observe the decrease and others—the increase in α-SNCA, depending on the type of the sample, time, and treatment duration [41,42,43]. Our data also add the type of mutation to the picture’s complexity. If we consider the results of marker-age correlation analysis also for GD patients, we could be entitled to formulate a hypothesis about the effect of ERT for GD on the mRNA level of α-synuclein. In patients of both GD types (i.e., those having the L444P variant and those carrying another *GBA1* variant), no correlation between the mRNA level and age was found. However, a strong age correlation was found for *GBA1* carriers, who obviously are not treated for GD: unknown *GBA1* variant carriers have mRNA levels that are negatively correlated with age, whereas L444P *GBA1* carriers accumulate the transcript material of α-synuclein with age. Probably the presence of the L444P variant in the *GBA1* gene can be a hint for searching for a further metabolic link between GD and an increased likelihood of PD. As proposed in the knockout mice model, L444P variant and accompanying α-synuclein accumulation leads the nigrostriatal dopamine neurons to an increased dopaminergic mitochondrial neurotoxin susceptibility, and thus plays dual physiological roles in the survival of the cells critically involved in parkinsonism [61]. Why the L444P variant upregulates *SNCA* mRNA expression with no clear link between *SNCA* mRNA transcript and total/oligomer α-SNCA protein level remains to be further studied.

## Figures and Tables

**Figure 1 biomolecules-13-00644-f001:**
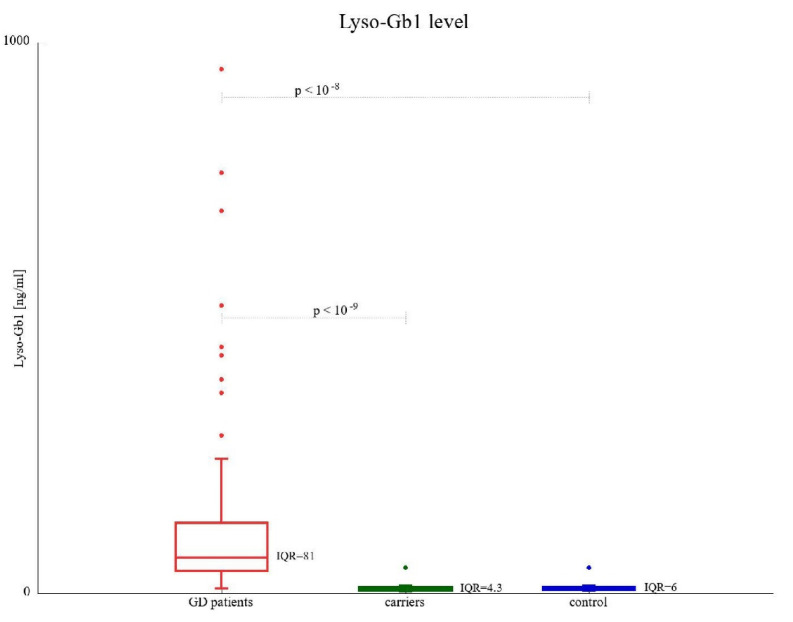
Lyso-Gb1 concentration (ng/mL) in GD patients and GBA1 carriers compared to healthy control subjects. IQR represents the interquartile range.

**Figure 2 biomolecules-13-00644-f002:**
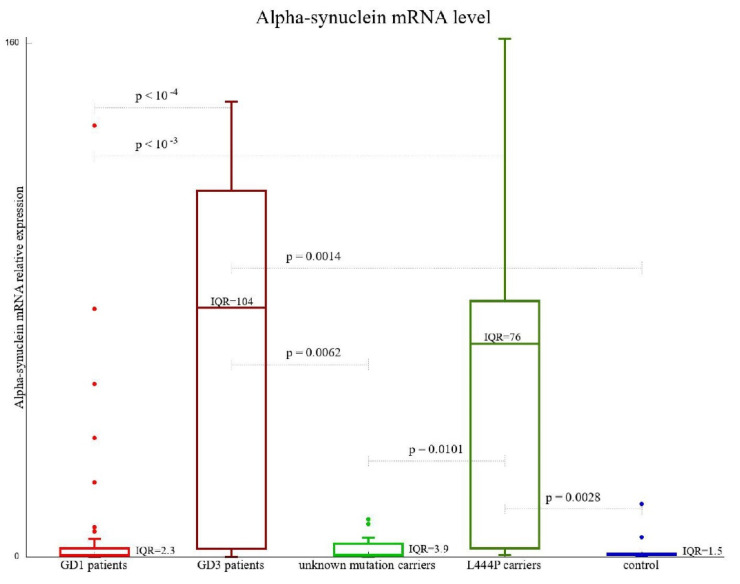
Alpha synuclein mRNA level. The *SNCA* gene expression level among GD1 and GD3 patients was compared to that among *GBA1* mutation carriers and healthy control subjects. Results are presented as the change-in-cycling-threshold using the ΔCq method.

**Figure 3 biomolecules-13-00644-f003:**
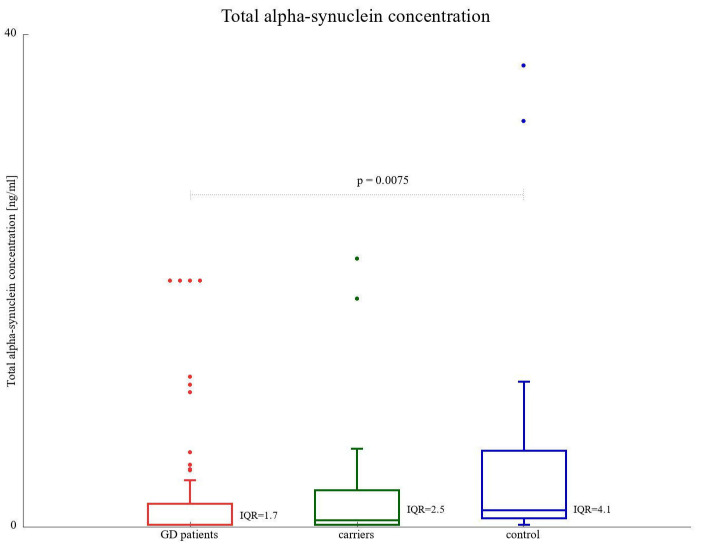
Total α-SNCA protein level (ng/mL) among GD1 and GD3 patients compared to GBA1 mutation carriers and healthy control subjects. IQR represents the interquartile range.

**Figure 4 biomolecules-13-00644-f004:**
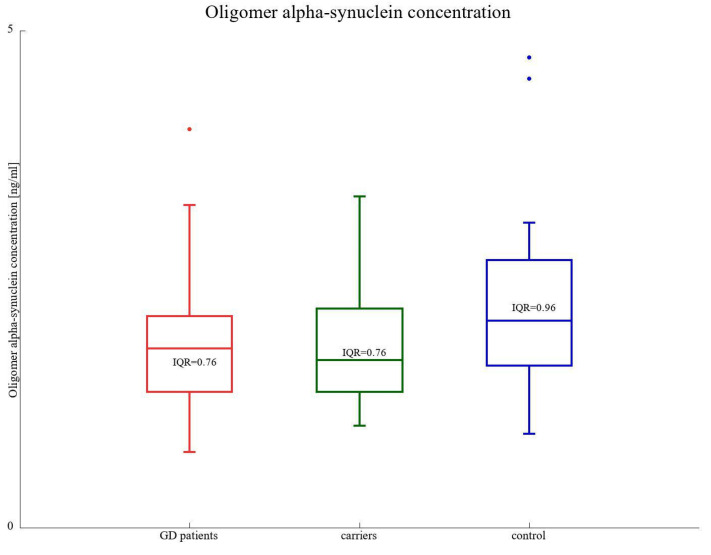
Oligomer α-SNCA protein concentration (ng/mL) among GD1 and GD3 patients compared to GBA1 mutation carriers and healthy control subjects. IQR represents the interquartile range.

**Table 1 biomolecules-13-00644-t001:** *GBA1* pathogenic variants in GD1 patients.

Number of Patients	Common Variant Name	DNA Nucleotide Change	Protein Change
28	N370S/L444P	c.1226A>G c.1448T>C	p.Asn409Ser p.Leu483Pro
6	N370S/ others	c.1226A>G	p.Asn409Ser
2	N370S/N370S	c.1226A>G	p.Asn409Ser
2	G377S/G377S	c.1246G>A	p.Gly416Ser
1	N370S/ G377S	c.1226A>G c.1246G>A	p.Asn409Ser p.Gly416Ser
1	N370S/84GG	c.1226A>G c.84dupG	p.Asn409Ser p.Leu29AlafsTer18
1	N370S/R496H	c.1226A>G c.1604G>A	p.Asn409Ser p.Arg535His
1	R433S/R433S	c.1416A>T	p.Arg444Ser
1	D438N/R87W	c.1312G>A c.259C>T	p.Asp438Asn p.Arg87Trp
1	D448G/R202X	c.1343A>G c.604C>T	p.Asp448Gly p.Arg202X
1	R48W/84GG	c.259C>T c.84dupG	p.Arg87Trp p.Leu29AlafsTer18
1	R48W/R48W	c.259C>T	p.Arg87Trp
1	84GG/ Other	c.84dupG	p.Leu29AlafsTer18

**Table 2 biomolecules-13-00644-t002:** Patients’ characteristics. Abbreviations: α-SNCA—alpha-synuclein; GD—Gaucher disease; Lyso-Gb1—Glucosylsphingosine; *SNCA*—gene coding alpha-synuclein.

	Healthy Controls	*GBA1* Mutation Carriers	GD1	GD3
*n*	16	19	47	18
Sex, F/M	10/6	14/5	28/19	12/8
Age in years, median (range)	44.5	52 (13–64)	40 (9–71)	30 (22–62)
Lyso-Gb1 (ng/mL), median (range)	8.49 (4.82–46.73)	7.51 (4.26–46.73)	71.5 (22.4–950.9)	64.55 (9.14–693.8)
*SNCA* mRNA level (2^−ΔcT^), median (range)	0.83 (0.08–16.38)	4.16 (0.02–161.42)	0.47 (0.01–134.42)	77.48 (0.04–141.72)
Total α-SNCA concentration (ng/mL), median (range)	1.34 (0.16–37.48)	0.53 (0.16–21.76)	0.16 (0.16–20)	0.92 (0.16–20)
Oligomer α-SNCA concentration (ng/mL), median (range)	2.08 (0.96–4.73)	1.69 (1.03–3.33)	1.76 (0.76–4.01)	1.857(0.76–3.03)

## Data Availability

All data generated or analysed during this study are included in this published article.

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
