# Peer review of "Alpha-Synuclein mRNA Level Found Dependent on L444P Variant in Carriers and Gaucher Disease Patients on Enzyme Replacement Therapy"

_biomolecules, 2023, doi:10.3390/biom13040644_

Round 1
Reviewer 1 Report
The paper reports interesting findings on the levels of synuclein and GlcSph but does not emphasise enough the most interesting results.
The fact that there is no correlation between GlcSph and synuclein in certain cohurts should be clear in the abstract and repeated at the end of the introduction and the beginning of the discussion. Instead it's buried in the detail of the paper. Clarity could be improved by more descriptive figure legends and titles that convey the bottom lines of the paper which are instead only in the discussion.
Please replace GluCer with the correct abbreviation GlcCer throughout. lyso-Gb1 should also be referred to as GlcSph.
Line 56 'the rain levels???
I don't think that the introduction emphasises the lack of GlcSph/GlcCer storage in heterozygote carriers or in mutations that are not associated with Gaucher E326K that are still associated with PD.
Author Response
Reviewer 1:
The paper reports interesting findings on the levels of synuclein and GlcSph but does not emphasise enough the most interesting results.
Answer: The authors are grateful for kind words.
Detailed comments:
- The fact that there is no correlation between GlcSph and synuclein in certain cohurts should be clear in the abstract and repeated at the end of the introduction and the beginning of the discussion. Instead it's buried in the detail of the paper. Clarity could be improved by more descriptive figure legends and titles that convey the bottom lines of the paper which are instead only in the discussion.
Answer: The authors are grateful for the comment and propose following changes:
- Figure legends were improved accordingly:
Page 5, line 191: Figure 1. Lyso-Gb1 concentration [ng/ml] in GD patients and GBA1 carriers compared to healthy control subjects. IQR represents interquartile range.
Page 6. Line 205: Figure 2. Alpha synuclein mRNA level. The SNCA gene expression level among GD1 and GD3 patients compared to GBA1 mutation carriers and healthy control subjects. Results are presented as the change-in-cycling-threshold using the ΔCq method.
Page 7 line 217: Figure 3. Total α-SNCA protein level [ng/ml] among GD1 and GD3 patients compared to GBA1 mutation carriers and healthy control subjects. IQR represents interquartile range.
Page 7, line 225: Figure 4. Oligomer α-SNCA protein concentration [ng/ml] among GD1 and GD3 patients compared to GBA1 mutation carriers and healthy control subjects. IQR represents interquartile range.
- Since the abstract is limited in the number of the words, we would like to keep current version. The authors believe that the most important results were briefly highlighted, namely:
α-synuclein mRNA level was found significantly elevated in GD3 patients and L444P carriers. GD1 patients, along with GBA1 carriers of an unknown or unconfirmed variant, as well as healthy controls, have the same low level of α-synuclein mRNA. There was no correlation found between the level of α-synuclein mRNA and age in GD patients treated with ERT, whereas there was a positive correlation in L444P carriers (Page 1, line 28). - Please replace GluCer with the correct abbreviation GlcCer throughout. lyso-Gb1 should also be referred to as GlcSph.
Answer: GluCer was replaced by GlcCer (Page 1, line 43). The authors would like to keep lyso-Gb1 as the more frequent used. All the results provided by the central labs are referred as the lyso-gb1 and therefore it is clearer for clinical readership.
- Line 56 'the rain levels??
Answer: The authors are sorry for the typo. It should be: “brain levels” (Page 2, line 56)
- I don't think that the introduction emphasises the lack of GlcSph/GlcCer storage in heterozygote carriers or in mutations that are not associated with Gaucher E326K that are still associated with PD.
Answer: the authors tried to be brief as expected by the journal and therefore just claimed in the previous version: Interestingly, carriers of GBA1 pathogenic variants are also posing a higher risk of developing PD (5, 30-31) (Page 2, line 89).
We added following sentence (Page 2, line 87): On the opposite, the glucocerobrosidase E326K variant is known to predispose to PD, but does not cause GD (29)
Reviewer 2 Report
The article “Alpha-synuclein mRNA level found dependent on L444P variant in carriers and Gaucher disease patients on enzyme replacement therapy” is a well-conceptualized study, emphasizing the role and involvement of alpha-synuclein in Gaucher disease and the changes associated with it in the case of enzyme replacement therapy. However, a few queries are mentioned below that, if addressed, can increase the overall scientific value of the study.
· What is the relevance of increased SNCA mRNA in GD3 patients and L444P mutation carriers when there’s no change observed in the synuclein protein or its oligomerization, which is the functional part attributing the disease phenotype?
· Is there any previous literature indicating higher levels of synuclein and its oligomers in healthy control subjects compared to GD patients and carriers? The data in figures 3 and 4 seem surprising. On the contrary, the mRNA levels of SNCA are lower in the control group (figure 2). What could be a possible explanation for that?
· Can the smaller cohort of subjects used in the study be responsible for contradictory results or non-reproducibility of data from previous studies? If so, do the authors plan on doing a bigger-scale study in the future?
· How reliable is the data, especially the changes described between groups, given the lack of age-matched subjects for different disease conditions?
· A few typographical errors were observed, like, SCNA instead of SNCA (line 99). A thorough proofread of the article is recommended.
Author Response
Reviewer 2:
The article “Alpha-synuclein mRNA level found dependent on L444P variant in carriers and Gaucher disease patients on enzyme replacement therapy” is a well-conceptualized study, emphasizing the role and involvement of alpha-synuclein in Gaucher disease and the changes associated with it in the case of enzyme replacement therapy. However, a few queries are mentioned below that, if addressed, can increase the overall scientific value of the study.
Answer: The authors are grateful for positive feedback and comments that help to increase the scientific value.
Detailed comments:
- What is the relevance of increased SNCA mRNA in GD3 patients and L444P mutation carriers when there’s no change observed in the synuclein protein or its oligomerization, which is the functional part attributing the disease phenotype?
Answer: The authors are grateful for this extremely important comment. Unfortunately, we can’t answer this question with current results and methodology we applied but we hope to follow up this question on the mice model in the next project. At this time point we can only speculate. Although, the elevated mRNA does not seem to be transcribed it could be that it plays some role in some cells or the protein is stacked and not detected by our experiments. As shown by Garcia-Esparcia et al., Alpha-synuclein expression and its oligomeryzation is extremely complex process. The authors found that altered machinery of protein synthesis is region- and stage-dependent and is associated with α-synuclein oligomers in Parkinson’s disease. Altered solubility and α-synuclein oligomer formation, assessed in total homogenate fractions blotted with anti-α-synuclein oligomer-specific antibody, was demonstrated by the authors in the substantia nigra and frontal cortex, but not in the putamen, in PD. Dramatic increase in α-synuclein oligomers was also seen in fluorescent-activated cell sorter (FACS)-isolated nuclei in the frontal cortex in PD.
Reference: Garcia-Esparcia, P., Hernández-Ortega, K., Koneti, A. et al. Altered machinery of protein synthesis is region- and stage-dependent and is associated with α-synuclein oligomers in Parkinson’s disease. acta neuropathol commun 3, 76 (2015). https://doi.org/10.1186/s40478-015-0257-4
- A) Is there any previous literature indicating higher levels of synuclein and its oligomers in healthy control subjects compared to GD patients and carriers? The data in figures 3 and 4 seem surprising. B) On the contrary, the mRNA levels of SNCA are lower in the control group (figure 2). What could be a possible explanation for that?
Answer: A) The authors are grateful for raising this important point. There is no previous literature describing higher levels of alpha-synuclein in healthy control subjects compared to GD patients and carriers.
- B) the disagreement in the transcription and the translation is one of the most mysterious finding of the research. There is no clear explanation and further studies are needed to clarify this. Perhaps, elevated lyso-gb1 level (carriers and gaucher patients vs healthy control) is involved in stacking alpha-synuclein in the cells or creation toxic oligo-protein complexes triggering dementia. There is not so much known on that but interesting paper was published by Mazzulli et al. The authors show on the cell culture that the bidirectional effect of a-syn and GCase forms a positive feedback loop that may lead to a self-propagating neurodegenerative disease. Our study is not in power to go so deep, since we evaluated samples from patients’ blood that could be limited by the nature.
Reference: Mazzulli JR, Xu YH, Sun Y, Knight AL, McLean PJ, Caldwell GA, Sidransky E, Grabowski GA, Krainc D. Gaucher disease glucocerebrosidase and α-synuclein form a bidirectional pathogenic loop in synucleinopathies. Cell. 2011 Jul 8;146(1):37-52. doi: 10.1016/j.cell.2011.06.001. Epub 2011 Jun 23. PMID: 21700325; PMCID: PMC3132082.
- Can the smaller cohort of subjects used in the study be responsible for contradictory results or non-reproducibility of data from previous studies? If so, do the authors plan on doing a bigger-scale study in the future?
Answer: The authors agree that the source of non-reproducible data could be the cohort size and its heterogeneity. What is important to be mentioned, our study on this topic (GD and associated markers of PD) is one of the biggest worldwide performed so far. Since Gaucher disease is rare disorder it is difficult to collect more patients. We described entire diagnosed polish population. We plan to extend the study to the international level as suggested by the Reviewer.
- How reliable is the data, especially the changes described between groups, given the lack of age-matched subjects for different disease conditions?
Answer: The authors agree that age-matched groups would better describe this health problem. Unfortunately, it is not possible for national study that we performed. As mentioned above, we collected entire diagnosed population. It is worthy to be mentioned that even clinical trials on VPRIV and Cerezyme (ERTs that are gold standard of the treatment for Gaucher disease) were performed on smaller than our populations i.e. VPRIV open label extension that pooled two clinical trials included 57 patients aged 3-62 years (Hughes et al).
Reference: Hughes DA, Gonzalez DE, Lukina EA, Mehta A, Kabra M, Elstein D, Kisinovsky I, Giraldo P, Bavdekar A, Hangartner TN, Wang N, Crombez E, Zimran A. Velaglucerase alfa (VPRIV) enzyme replacement therapy in patients with Gaucher disease: Long-term data from phase III clinical trials. Am J Hematol. 2015 Jul;90(7):584-91. doi: 10.1002/ajh.24012. PMID: 25801797; PMCID: PMC4654249.
- A few typographical errors were observed, like, SCNA instead of SNCA (line 99). A thorough proofread of the article is recommended.
Answer: The authors went through the article and corrected number of typos including line 99.
Reviewer 3 Report
This manuscript aims to better clarify the link between Gaucher disease (GD) and Parkinson’s disease (PD), as mutation in the β-glucocerebrosidase GBA1 both drives GD and is one of the most important risk factors for the development of PD. It is thought that expression of GBA1 mutants drives α-synuclein aggregation, a predicted driving force of PD. GD has three distinct clinical forms GD1-3, where GD1 is the most common and is not believed to cause neurological damage. GD2 and GD3 are both characterized by progressive neurological impairment. There are two main mutations in GBA1 found in GD patients, L444P and N370S, though it was not clear if these mutations are specific to one of the clinical forms.
Here, the authors measure whether GD1 and GD3 patients (not clear why GD2 patients were left out) have higher levels of α-synuclein mRNA expression, protein expression, or oligomer formation. They find that GD3 patients and L444P carriers have higher levels of α-synuclein mRNA, but there are no differences in protein levels or oligomerization.
I do not recommend this manuscript for publication, as the overall findings are mostly negative and provide very incremental insight into the field. The clinical relevance of increased mRNA, but not protein levels or oligomerization, is unclear, as it is believed increased levels of α-synuclein protein aggregates is what drives PD. It also seems that this manuscript is unable to reproduce other well-established data in the field, including an increase in Lyso-Gb1 levels between GD1 and GD3 patients. While some explanations are given for this, it casts doubt on how reliable the data and statistical analysis are.
Author Response
Reviewer 3.
This manuscript aims to better clarify the link between Gaucher disease (GD) and Parkinson’s disease (PD), as mutation in the β-glucocerebrosidase GBA1 both drives GD and is one of the most important risk factors for the development of PD. It is thought that expression of GBA1 mutants drives α-synuclein aggregation, a predicted driving force of PD. GD has three distinct clinical forms GD1-3, where GD1 is the most common and is not believed to cause neurological damage. GD2 and GD3 are both characterized by progressive neurological impairment. There are two main mutations in GBA1 found in GD patients, L444P and N370S, though it was not clear if these mutations are specific to one of the clinical forms.
Answer: The authors are grateful for the comments provided by the Reviewer. Since our manuscript is not aiming to describe GD clinical forms itself, the authors do not discuss the pathogenic variants and its impact on the triggering certain form. We briefly described our cohort (page 4, line 166):
65 Polish patients with GD, 47 with type 1 and 18 with type 3, were enrolled into the study. All the patients displayed a decreased activity of β-glucocerebrosidase, combined with genetic confirmation (detection of GBA1 biallelic pathogenic variants). Regarding patients’ genotype, the entire GD3 population comprised of L444P homozygotes, and GD1 patients presented various variant, see Table 1.
Detailed comments:
- Here, the authors measure whether GD1 and GD3 patients (not clear why GD2 patients were left out) have higher levels of α-synuclein mRNA expression, protein expression, or oligomer formation. They find that GD3 patients and L444P carriers have higher levels of α-synuclein mRNA, but there are no differences in protein levels or oligomerization. (Page 4 line 66)
Answer: The authors would like to explain why GD2 patients are not in the scope of the study. Type 2 Gaucher disease is a very rare, rapildly progessive form of Gaucher disease which affects the brain (central nervous system) as well as the spleen, liver, lungs and bones. Formerly called infantile Gaucher disease, it is characterised by severe neurological (brain) involvement in the first year of life. Most affected children die before age 2.
Our manuscript described the results we obtained. It is valid to discuss the relevance of transcript vs protein level in this unique cohort. Regarding alpha-synuclein itself, contradictory results were previously published as covered in perfect review by Ganguly et al.
We believe our findings add to the discussion and help to understand heterogenicity observed in the clinics among Gaucher patients.
Reference: Ganguly U, Singh S, Pal S, Prasad S, Agrawal BK, Saini RV, Chakrabarti S. Alpha-Synuclein as a Biomarker of Parkinson's Disease: Good, but Not Good Enough. Front Aging Neurosci. 2021 Jul 8;13:702639. doi: 10.3389/fnagi.2021.702639. PMID: 34305577; PMCID: PMC8298029.
- I do not recommend this manuscript for publication, as the overall findings are mostly negative and provide very incremental insight into the field. The clinical relevance of increased mRNA, but not protein levels or oligomerization, is unclear, as it is believed increased levels of α-synuclein protein aggregates is what drives PD. It also seems that this manuscript is unable to reproduce other well-established data in the field, including an increase in Lyso-Gb1 levels between GD1 and GD3 patients. While some explanations are given for this, it casts doubt on how reliable the data and statistical analysis are.
Answer: The authors are grateful for the critical review but strongly disagree with this comment. There are certain strengths of the paper we would like to highlight:
- Relatively big cohort was included (considering rare disease). Our efforts bring entire polish diagnosed population. The study is one of the biggest of its kind and collected cohort bigger than pooled analysis of the ERT clinical trials
- We involved both GD1 and GD3 patients what is unique and never studied before
- We captured all the patients and carriers in the same time covering number of molecular techniques that is covering entire picture: from mutations triggering GD and GD biomarker to alpha-synuclein transcript and protein/oligomer level
- Regarding the lyso-gb1 levels in GD1 and GD3 patients. This biomarker was measured independent by the central laboratory (Archimed Vienna) that covers clinical trials in gaucher disease field and is the diagnostic partner of Genzyme – top rare disease pharma company. We believe it is reliable and results from this laboratory are cited worldwide.
Reviewer 4 Report
The manuscript “Alpha-synuclein mRNA level found dependent on L444P vari-2 ant in carriers and Gaucher disease patients on enzyme replacement therapy” by Dubiela and co-workers relies with Gaucher disease (GD), a rare genetic disease, and Parkinson disease (PD), one of the most common neurodegenerative diseases.
The authors evaluated the contents of glucosylsphingosine (Lyso-Gb1) and α-synuclein (the main biomarkers for GD and PD respectively) in a cohort of patients comprising 65 patients with GD treated with ERT (47 GD1 patients and 18 GD3 patients), 19 GBA1 pathogenic variants carriers (including 10 L444P carriers) and 16 healthy subjects. The main goal was to highlight possible correlations between the biomarkers and specific subgroups of patients.
They found statistically relevant differences regarding α-synuclein mRNA level in GD3 patients and L444P carriers compared to controls, GD1 and unknown mutation carriers. They also investigated possible correlations with age.
Overall, the paper is clearly written and provides useful results that could shed light on both diseases etiology, their molecular mechanisms, and as a consequence could lead to the development of better personalised therapeutic approaches.
I think that the manuscript deserves to be published.
I would suggest to improve the quality of the figures in terms of colours and dimensions of symbols and characters. This refers to the SupplFig2 too. I also suggest to add a legend to the SupplFig2.
Author Response
Reviewer 4:
The manuscript “Alpha-synuclein mRNA level found dependent on L444P vari-2 ant in carriers and Gaucher disease patients on enzyme replacement therapy” by Dubiela and co-workers relies with Gaucher disease (GD), a rare genetic disease, and Parkinson disease (PD), one of the most common neurodegenerative diseases.
The authors evaluated the contents of glucosylsphingosine (Lyso-Gb1) and α-synuclein (the main biomarkers for GD and PD respectively) in a cohort of patients comprising 65 patients with GD treated with ERT (47 GD1 patients and 18 GD3 patients), 19 GBA1 pathogenic variants carriers (including 10 L444P carriers) and 16 healthy subjects. The main goal was to highlight possible correlations between the biomarkers and specific subgroups of patients.
They found statistically relevant differences regarding α-synuclein mRNA level in GD3 patients and L444P carriers compared to controls, GD1 and unknown mutation carriers. They also investigated possible correlations with age.
Overall, the paper is clearly written and provides useful results that could shed light on both diseases etiology, their molecular mechanisms, and as a consequence could lead to the development of better personalised therapeutic approaches.
I think that the manuscript deserves to be published.
Answer: The authors are grateful for positive feedback
I would suggest to improve the quality of the figures in terms of colours and dimensions of symbols and characters. This refers to the SupplFig2 too. I also suggest to add a legend to the SupplFig2.
Answer: The quality of the figures and the way of graphical representation will be improved by the authors after manuscript acceptance according to the Journal requirements. The project was performed without any financial support beside experimental investments. All the figures were prepared on the free licence software. Once the publication will be accepted, we will get the funding for the improvement
The legend was added: (Suppl. Material). There was a mistake in numbering Suppl Material - now it is a Suppl Table 1 and Suppl Figure 1
Suppl. Fig 1. Scatter plot of age (x-axis) and biomarkers (Lyso-Gb1, α-SNCA mRNA level, α-SNCA total protein concentration and oligomer concentration). Different point types indicate different subgroups.
Suppl. Table 1. Pearson linear correlation coefficient between the given marker (Lyso- Gb1, SNCA mRNA level, α-SNCA total protein concentration and oligomer concentration) and age in the following groups: GD1, GD3, unknown mutation carriers, L444P carriers, and control.
Round 2
Reviewer 3 Report
The explanations of which variants correspond to which clinical form and why GD2 patients were not used was helpful. However, the response to primarily negative data was not sufficient. I understand that it was a large and varied cohort that was used, but it needs to be better explained why changes in only alpha-synuclein mRNA could be clinically important in the absence of changes in protein levels or oligomerization.
Author Response
The authors are grateful for positive feedback and the possibility to further discuss our findings. We fully agree that our results might seem somehow surprising but we believe this is a typical scientific headache – straightforward findings do not explain the complexity of phenomena. Although the elevated mRNA that does not seem to be transcribed could play a role in some cells or the protein is stacked and not detected by our experiments. As shown by Garcia-Esparcia et al., Alpha-synuclein expression and its oligomeryzation is an extremely complex process. The authors found that altered machinery of protein synthesis is region- and stage-dependent and is associated with α-synuclein oligomers in Parkinson’s disease. Altered solubility and α-synuclein oligomer formation, assessed in total homogenate fractions blotted with anti-α-synuclein oligomer-specific antibody was demonstrated by the authors in the substantia nigra and frontal cortex, but not in the putamen, in PD. Dramatic increase in α-synuclein oligomers was also seen in fluorescent-activated cell sorter (FACS)-isolated nuclei in the frontal cortex in PD.(1) Moreover, there are plenty of well performed studies describing similar phenomena. As summarized by Nie et al. there are three potential reasons for the lack of strong correlation between mRNA and protein expression levels: (i) translational regulation, (ii) differences in protein in vivo half-lives, and (iii) the significant amount of experimental error, including differences in experimental conditions. All of them could play a role, but they do not discriminate clinical relevance of our finding that so far is the biggest of its kind. We would therefore like to share our data with the readership to open further discussion on the clinical importance of the finding.